# Consistency Regularization for Certified Robustness of Smoothed Classifiers

**Jongheon Jeong**[1]    **Jinwoo Shin**[2,1]
[1]School of Electrical Engineering    [2]Graduate School of AI
Korea Advanced Institute of Science and Technology (KAIST)
Daejeon, South Korea
{jongheonj, jinwoos}@kaist.ac.kr

## Abstract

A recent technique of *randomized smoothing* has shown that the worst-case (adversarial) $\ell_2$-robustness can be transformed into the average-case Gaussian-robustness by "smoothing" a classifier, i.e., by considering the averaged prediction over Gaussian noise. In this paradigm, one should rethink the notion of adversarial robustness in terms of generalization ability of a classifier under noisy observations. We found that the trade-off between accuracy and certified robustness of smoothed classifiers can be greatly controlled by simply regularizing the prediction consistency over noise. This relationship allows us to design a robust training objective without approximating a non-existing smoothed classifier, e.g., via soft smoothing. Our experiments under various deep neural network architectures and datasets show that the "certified" $\ell_2$-robustness can be dramatically improved with the proposed regularization, even achieving better or comparable results to the state-of-the-art approaches with significantly less training costs and hyperparameters.

## 1 Introduction

Despite achieving even super-human level performance on *i.i.d.* datasets [16, 33, 11], deep neural network (DNN) classifiers usually make substantially fragile predictions than humans on the samples not from the data-generating distribution. The broad existence of *adversarial examples* [35, 15] are arguably the most crucial instance of this phenomenon: a small, adversarially-crafted perturbation on input can easily change the prediction of a classifier, even when the perturbation does not affect the semantic information perceived by humans at all.

This intriguing weakness of DNNs has encouraged many researchers to develop *robust* neural networks, along with a parallel attempt to break them with stronger attacks [6, 38, 2]. Currently, the community has agreed that *adversarial training* [15, 24, 45], i.e., augmenting the training dataset with adversarial examples, is an effective defense method, but the "scalability" of the method is often questionable in several aspects: (a) it is generally hard to guarantee that an adversarially-trained classifier is indeed robust, (b) generalizing the robustness beyond the training threat model is still challenging [36, 19], and (c) the network capacity required for robust representation seems to be much larger than practice, e.g., a recent observation shows empirical robustness does not saturate even at ResNet-638 on ImageNet dataset [41].

Alternatively, a growing body of the research has developed methods that can provide *certified robustness* [34, 39, 46]. *Randomized smoothing* [22, 10] is a recent idea in this direction, which shows that any classifier (e.g., a neural network) that performs well under Gaussian noise can be "smoothed" into a certifiably robust classifier. This opens a new, scalable notion of adversarial robustness: a neural network may not have to be perfectly smooth, as a proxy of another classifier.

However, it has been relatively under-explored that how to train a good base classifier to maximize the certified robustness of the smoothed counterpart, e.g., Cohen et al. [10] only explored the standard training with Gaussian augmentation. A few recent works [32, 44] have shown that a more sophisticated training algorithm can indeed improve the certified robustness, but the common downside is that they require a sensitive choice of many hyperparameters to optimally trade-off between accuracy and robustness, often imposing a significant amount of additional training costs.

**Contribution.**    In this paper, we show that a simple *consistency regularization* term added on a standard training scheme surprisingly improves the certified robustness of smoothed classifiers. Maintaining the prediction consistency over a certain noise, e.g., Gaussian, can be regarded as a natural and desirable property for a classifier under noisy observations. Indeed, for example, forcing such consistency is now considered as one of the most popular techniques in the semi-supervised learning literature [31, 25, 27, 3]. We examine this regularization, motivated by the observation that perfect consistency is a sufficient condition for minimizing the robust 0-1 loss of smoothed classifiers. This observation connects certified robustness of smoothed classifiers to the general corruption robustness [17, 14], supporting a great potential of smoothed inference as a scalable alternative of adversarially-trained, deterministic classifiers.

We verify the effectiveness of our proposed regularization based on extensive evaluation covering MNIST [21], CIFAR-10 [20], and ImageNet [30] classification datasets. We show that our simple technique upon a naïve training achieves a very comparable, or even better, certified $\ell_2$-robustness to other recent, robust training methods [23, 32, 44]. For example, one of our models for CIFAR-10 shows a better robustness than those trained by other tested methods with $2.7\times$ faster training due to its simplicity. Furthermore, we also demonstrate that applying our method upon a more sophisticated training even further improves the certified robustness, e.g., our method applied upon state-of-the-art training could further improve the average certified $\ell_2$-radius $0.785 \rightarrow 0.816$ on CIFAR-10.

Despite its effectiveness, our proposed regularization is easy-to-use with fewer hyperparameters, and could run significantly faster than existing approaches without additional backward computation as in adversarial training. We observe that our method does not introduce instability in training for a wide range of hyperparameters, offering a new, stable trade-off term between accuracy and certified robustness of smoothed classifiers. Finally, our concept of regularizing prediction consistency can be extended to other families of noise other than Gaussian, which are often corresponded to different types of adversary, e.g., Laplace noise for $\ell_1$-robustness [22, 13, 42].

## 2   Preliminaries

### 2.1   Adversarial robustness

We consider a classification task with $K$ classes from a dataset $\mathcal{D} = \{(x_i, y_i)\}_{i=1}^n$, where $x \in \mathbb{R}^d$ and $y \in \mathcal{Y} := \{1, \cdots, K\}$ denote an input and the corresponding class label, respectively. Usually, $\mathcal{D}$ is assumed to be *i.i.d.* samples from a data-generating distribution $P$. Let $f : \mathbb{R}^d \rightarrow \mathcal{Y}$ be a classifier. In many cases, e.g., neural networks, this mapping is modeled by $f(x) := \arg\max_{k \in \mathcal{Y}} F_k(x)$ with a differentiable mapping $F : \mathbb{R}^d \rightarrow \Delta^{K-1}$ for a gradient-based optimization, where $\Delta^{K-1}$ denotes the probability simplex in $\mathbb{R}^K$.

In the literature of general robustness research [12, 5, 17], $f$ is required to perform well not only on $P$, but also on a certain extension of it without changing the semantics, say $\widetilde{P}$. In particular, the notion of *adversarial robustness* considers the worst-case distribution near $P$ under a certain distance metric. More concretely, a common way to define the adversarial robustness is to consider the *average minimum-distance* of adversarial perturbation [26, 7, 8], namely:

$$R(f; P) := \mathbb{E}_{(x,y)\sim P} \left[ \min_{f(x') \neq y} ||x' - x||_2 \right]. \tag{1}$$

Therefore, our goal is to train $f$ that (a) performs well on $P$, while (b) maximizing $R(f; P)$ as well.

### 2.2   Randomized smoothing

In practice, the inner minimization objective in (1) is usually not easy to optimize exactly, and mostly results in near-zero value on standard neural network classifiers. The key idea of *randomized*

*smoothing* [10] is rather to consider the robustness of a "smoothed" transformation of the *base classifier* $f$ over Gaussian noise, namely $\hat{f}$:

$$\hat{f}(x) := \underset{k \in \mathcal{Y}}{\arg\max} \, \mathbb{E}_{\delta \sim \mathcal{N}(0, \sigma^2 I)} \left[ \mathbf{1}_{f(x+\delta)=k} \right], \tag{2}$$

where $\mathbf{1}_A$ denotes the *indicator* random variable, formally defined by $\mathbf{1}_A(\omega) = 1$ if $\omega \in A$ and 0 otherwise, and $\sigma^2$ is a hyperparameter that controls the level of smoothing. For a given input $x$, Cohen et al. [10] guarantees a *certified radius* in $\ell_2$ distance, the current state-of-the-art lower bound of the minimum-distance of adversarial perturbation around $\hat{f}(x)$: suppose $f(x + \delta)$ returns a class $\hat{f}(x) \in \mathcal{Y}$ with probability $p^{(1)}$ and the "runner-up" (i.e., the second best) class with probability $p^{(2)} := \max_{c \neq \hat{f}(x)} \mathbb{P}(f(x + \delta) = c)$. Then, the lower bound can be given as follows:

$$R(\hat{f}; x, y) := \min_{\hat{f}(x') \neq y} ||x' - x||_2 \geq \frac{\sigma}{2} \left( \Phi^{-1}(p^{(1)}) - \Phi^{-1}(p^{(2)}) \right) \tag{3}$$

provided that $\hat{f}(x) = y$, otherwise $R(\hat{f}; x, y) := 0$. Here, $\Phi$ denotes the standard Gaussian CDF. Since the inequality holds for any upper bound of $p^{(2)}$, say $\overline{p^{(2)}}$, one could also obtain a bit loose, but simpler bound of certified radius by letting $\overline{p^{(2)}} = 1 - p^{(1)} \geq p^{(2)}$:

$$R(\hat{f}; x, y) \geq \sigma \cdot \Phi^{-1}(p^{(1)}) =: \underline{R}(\hat{f}, x, y). \tag{4}$$

## 3 Consistency regularization for smoothed classifiers

Our intuition on the proposed consistency regularization is based on minimizing the *0-1 robust classification loss*, in a similar manner to the recent attempts of decomposing the training objective with respect to the accuracy and robustness [45, 44]. Specifically, we attempt minimize the following:

$$\mathbb{E}_{(x,y) \in \mathcal{D}} \left[ 1 - \mathbf{1}_{\underline{R}(\hat{f}; x, y) \geq \varepsilon} \right] = \underbrace{\mathbb{E} \left[ \mathbf{1}_{\hat{f}(x) \neq y} \right]}_{\text{natural error}} + \underbrace{\mathbb{E} \left[ \mathbf{1}_{\hat{f}(x) = y, \, \underline{R}(\hat{f}; x, y) < \varepsilon} \right]}_{\text{robust error}}, \tag{5}$$

where $\underline{R}$ is the certified lower bound of $R$ as defined in (4), and $\varepsilon > 0$ is a pre-defined constant. Assuming that the natural error term can be optimized via a standard surrogate loss, e.g., cross-entropy, we rather focus on how to minimize the robust error term. Here, the key difficulties to consider a gradient-based optimization is that (a) computing $\hat{f}$ exactly is intractable, and more importantly, (b) $\hat{f}$ is practically a non-differentiable object when estimated via Monte Carlo sampling (see (2)), so that even a proper surrogate loss function would not make the optimization differentiable.

To bypass these issues, we instead concentrate on a *sufficient condition* to minimize the given robust 0-1 loss. Recall that we assume $f(x) = \arg\max_{k \in \mathcal{Y}} F(x)$ for a differentiable function $F: \mathbb{R}^d \to \Delta^{K-1}$. Here, we notice that the robust loss in (5) would anyway become zero if $F(x + \delta)$ returns a *constant* output over $\delta$ for a given $x$. Indeed, this implies $\mathbb{P}_\delta(f(x + \delta) = \hat{f}(x))$ to become 1 *regardless* of what $\hat{f}$ is, and minimizes an upper bound of the robust loss in (5) due to the following:

$$\mathbb{E}_{(x,y) \in \mathcal{D}} \left[ \mathbf{1}_{\hat{f}(x) = y, \, \underline{R}(\hat{f}; x, y) < \varepsilon} \right] = \mathbb{E} \left[ \mathbf{1}_{\hat{f}(x) = y, \, \underline{R}(\hat{f}; x, \hat{f}(x)) < \varepsilon} \right]$$
$$\leq \mathbb{E} \left[ \mathbf{1}_{\underline{R}(\hat{f}; x, \hat{f}(x)) < \varepsilon} \right] = \mathbb{E} \left[ \mathbf{1}_{\mathbb{P}_\delta(f(x+\delta)=\hat{f}(x)) < \Phi(\frac{\varepsilon}{\sigma})} \right], \tag{6}$$

where the last equality is from the definition of $\underline{R}$ in (4). Therefore, we attempt to optimize the robust training objective on $\hat{f}$ via regularizing $F(x + \delta)$ to be *consistent* across $\delta$. Specifically, we propose the following consistency regularization upon any standard training objective:

$$L^{\text{con}} := \lambda \cdot \mathbb{E}_\delta \left[ \text{KL}(\hat{F}(x) || F(x + \delta)) \right] + \eta \cdot \text{H}(\hat{F}(x)), \tag{7}$$

where $\hat{F}(x) := \mathbb{E}[F(x + \delta)]$ is the mean of $F(x + \delta)$, $\text{KL}(\cdot || \cdot)$ and $\text{H}(\cdot)$ denote the Kullback–Leibler (KL) divergence and the entropy, respectively, and $\lambda, \eta > 0$ are hyperparameters that control the relative strength. In other words, this regularization enforces $F$, correspondingly $f$ as well, to reduce the *variance* of predictions under Gaussian noise for a given sample $x$, while preventing the mean

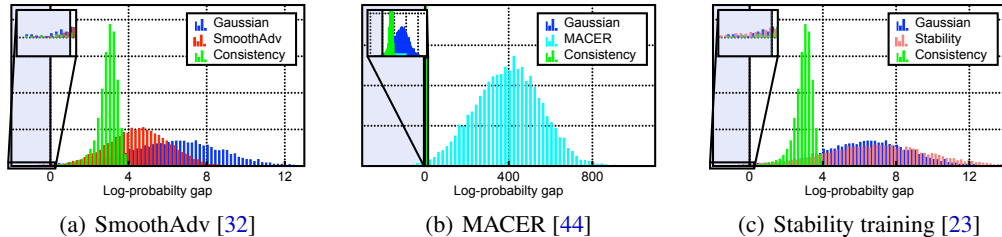

(a) SmoothAdv [32]  (b) MACER [44]  (c) Stability training [23]

Figure 1: Comparison of log-probabilty distributions under Gaussian noise at a fixed test sample of MNIST. For each histogram, we use 10,000 samples of noise. "Gaussian" indicates the baseline training with Gaussian augmentation [10], and "Consistency" indicates our proposed regularization applied upon "Gaussian". The left, shaded areas are where a classifier makes a misclassification.

to be too close to the uniform via the entropy penalty. Note that the proposed form (7) includes the cross-entropy loss $\mathbb{E}[\mathcal{L}(F(x+\delta), \hat{F}(x))]$ when $\lambda = \eta$. In practice, we observe $\lambda$ plays a more crucial role than $\eta$ for the trade-off between accuracy and robustness: e.g., in our experiments, we use a fixed $\eta = 0.5$ unless otherwise noted, and adjust $\lambda$ to control the robustness.

We also remark that, for a fixed $x$, (7) gives a family of *calibrated* [28, 45] surrogate losses of the 0-1 risk $\mathbb{E}_\delta[\mathbf{1}_{f(x+\delta) \neq \hat{f}(x)}] = \mathbb{P}_\delta(f(x+\delta) \neq \hat{f}(x))$, i.e., minimizers of (7) are also those of the 0-1 risk, and thus it minimizes the upper bound in (6) when minimized across $(x, y) \sim \mathcal{D}$. Similarly, one can adopt other forms of consistency regularization as long as the regularization leads (6) to be zero: we examine such variants in Section 4.5, and it turns out indeed they are also effective to improve the certified robustness, while our form (7) shows a particular robustness compared to them empirically.

### 3.1 Comparison to prior works

There have been a few prior approaches in attempts to improve the robustness of smoothed classifiers with a more sophisticated training method beyond that of Cohen et al. [10]. Salman et al. [32] proposed *SmoothAdv*, which shows that adversarial training directly on smoothed classifiers improve the certified robustness. More recently, Zhai et al. [44] proposed a faster training method called *MACER*, via maximizing a soft approximation of the certified radius given in (3).[1] The essential difference of our regularization to the previous works is at how the non-differentiable $\hat{f}$ is handled, namely, prior works commonly approximate $\hat{f}$ directly by the inner soft-classifier $F$:

$$\mathbb{E}_\delta[\mathbf{1}_{f(x+\delta)=k}] \approx \mathbb{E}_\delta[F_k(x+\delta)], \tag{8}$$

for each class $k \in \mathcal{Y}$. A key caveat here is that, however, optimizing $\hat{f}$ with this approximation would implicitly count out much optimal solutions of $F$. More specifically, we remark that an optimal soft classifier $F$ does not require to have confidence near to 1 for maximizing the certified radius (3), which is a usual solution found by minimizing the cross-entropy based on (8). Our approach rather considers an "indirect" regularizer of $\hat{f}$ without assuming such an approximation, thereby allows a more flexible optimization.

On the other hand, Li et al. [23] proposed *stability training*, as a parallel attempt to the Gaussian training of randomized smoothing [10] to obtain a robust smoothed classifier: namely, in order to perform well on Gaussian noise, stability training trains $F$ with the following loss:

$$\min_F \mathcal{L}(F(x), y) + \lambda \cdot \mathcal{L}(F(x), F(x+\delta)), \tag{9}$$

where $\mathcal{L}$ is the cross-entropy loss. The regularization term used in (9) has a seemingly similar formula to ours particularly when $\lambda = \eta$ in (7), but there is a fundamental difference: our method (7) does not require $F$ to minimize $\mathcal{L}(F(x), y)$ to perform well on $(x+\delta, y)$. Consequently, our method again allows a more flexible solution compared to (9). In Section 4.2, we empirically show that our form of regularization (7) attains a significantly better robustness than (9). We also show in the supplementary material that, due to such flexibility, our method is much more robust on the choice of $\lambda$.

**Log-probability gap.** In Figure 1, we illustrate how the optimal classifier found by our method differs from others, by comparing the distribution of *log-probability gap* over Gaussian noise $\delta$ for a given (noisy) sample $(x + \delta, y)$, namely $\log F_y(x + \delta) - \max_{c \neq y} \log F_c(x + \delta)$. In other words, we compare the *output margin* of $f$ at $(x + \delta, y)$ to observe the *input margin* of $\hat{f}$: the robustness guarantee in (3) implies that it is enough to minimize $\mathbb{P}_\delta(f(x + \delta) \neq y)$ to improve the robustness of $\hat{f}$ at $x$. This can be also viewed under the Lipschitzness angle: Salman et al. [32] show that any Gaussian-smoothed classifier $\hat{f}$ has an explicit Lipschitz constant, leading to a simpler proof of (3).

Overall, we observe in Figure 1 that our method learns relatively lower, yet more consistent, confidences than others. We also found that MACER [44] tends to vary on much larger values in logits (see Figure 1(b)): MACER essentially maximizes the gap between the first- and second-best logits of $\mathbb{E}[F(x + \delta)]$, which leads $F$ to have an arbitrary large value when optimized. Finally, Figure 1(c) supports that our method is fundamentally different to the stability training [23]: one can observe that stability training does not give a particular consistency that our method shows.

## 3.2 Training with consistency regularization

**Overall training objective.** Combining our regularization $L^{\mathtt{con}}$ to a natural surrogate loss $L^{\mathtt{nat}}$ leads to a full objective to minimize. Any form of $L^{\mathtt{nat}}$ is possible to use, as long as it minimizes the natural error of $\hat{f}$ in (5), e.g., the standard cross-entropy loss on $f$ may not be proper for $L^{\mathtt{nat}}$. As a plain example, we use the loss proposed by Cohen et al. [10], which simply performs Gaussian augmentation during training: for a given sample $(x, y) \sim \mathcal{D}$, the authors suggest to minimize:

$$L^{\mathtt{nat}} := \mathbb{E}_{\delta \sim \mathcal{N}(0, \sigma^2 I)} \left[ \mathcal{L}(F(x + \delta), y) \right]. \tag{10}$$

The overall objective with consistency regularization is then:

$$L := L^{\mathtt{nat}} + L^{\mathtt{con}} = \mathbb{E}_\delta \left[ \mathcal{L}(F(x + \delta), y) + \lambda \cdot \mathrm{KL}(\hat{F}(x) || F(x + \delta)) + \eta \cdot \mathrm{H}(\hat{F}(x)) \right] \tag{11}$$

$$\approx \frac{1}{m} \sum_i \left( \mathcal{L}(F(x + \delta_i), y) + \lambda \cdot \mathrm{KL}(\hat{F}(x) || F(x + \delta_i)) \right) + \eta \cdot \mathrm{H}(\hat{F}(x)), \tag{12}$$

where (12) is a concrete loss of (11) via Monte Carlo sampling over $\delta$. Nevertheless, our regularization scheme is not limited to a specific choice of $L^{\mathtt{nat}}$, and one can also apply others in a similar way. In our experiments, for example, we show that using SmoothAdv [32] as $L^{\mathtt{nat}}$ could further improve the certified robustness, although it is significantly more expensive to optimize compared to (10).

**Computational overhead.** We use $m$ independent samples of Gaussian noise in (12) to estimate (11), and this is the only source of extra training costs compared to the original [10]. Nevertheless, we empirically observe that the minimal choice[2] of $m = 2$ is fairly enough for our method, as demonstrated in Figure 3(b). Considering that other existing methods also use this inner-sampling procedure, often requiring an additional outer-loop of backward computations for adversarial training [32], or a large number of $m$ for a stable training [44], our method offers a significantly less training cost with less hyperparameters, as further discussed in Section 4.4.

## 4 Experiments

We validate the effectiveness of our proposed consistency regularization for a wide range of image classification datasets: MNIST[3] [21], CIFAR-10 [20], and ImageNet [30].[4] Overall, our results consistently demonstrate that simply applying our method in addition to the other baseline training methods greatly boosts the certified $\ell_2$-robustness via randomized smoothing. Remarkably, we show that our method even further improves the previous state-of-the-art results of SmoothAdv [32]. We also perform an ablation study to further investigate the detailed components proposed in our method.

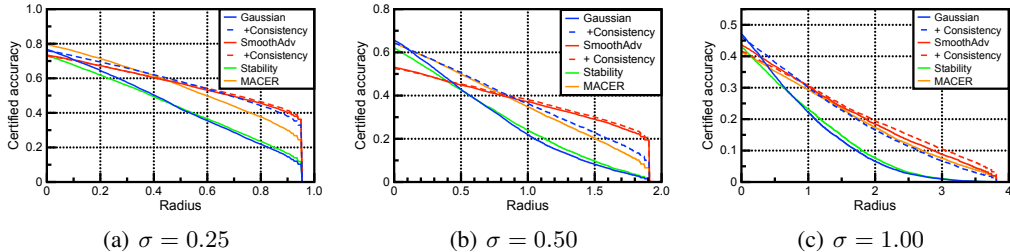

|  (a) $\sigma = 0.25$ | (b) $\sigma = 0.50$ | (c) $\sigma = 1.00$ |

Figure 2: Comparison of approximate certified accuracy via randomized smoothing for various training methods on CIFAR-10. A sharp drop of certified accuracy in the plots exists since there is a hard upper bound that CERTIFY can output for a given $\sigma$ and $n = 100,000$.

## 4.1 Setups

**Evaluation metrics.** To evaluate certified robustness for a given classifier $f$, we aim to compute the *certified test accuracy* at radius $r$, which is defined by the fraction of the test dataset that $\hat{f}$ can certify the robustness of radius $r$ with respect to the certifiable lower bound in (3). Due to the intractability of this metric, however, we instead measure the *approximate certified test accuracy* [10]. More concretely, Cohen et al. [10] proposed a practical Monte Carlo based certification procedure, namely CERTIFY, which returns the prediction of $\hat{f}$ and a "safe" lower bound of certified radius over the randomness of $n$ samples with probability at least $1 - \alpha$, or abstains the certification. The *approximate certified test accuracy* is then defined by the fraction of the test dataset which CERTIFY classifies correctly with radius larger than $r$ without abstaining.

In our experiments, we use the official implementation[5] of CERTIFY for evaluation, with $n = 100,000$, $n_0 = 100$ and $\alpha = 0.001$, following prior works [10, 32]. We mainly report the approximate certified test accuracy at various radii, but also report the *average certified radius* (ACR) considered by Zhai et al. [44], i.e., the averaged value of certified radii returned by CERTIFY, as another metric for better comparison of robustness under the trade-offs between accuracy and robustness [37, 45], namely $\mathrm{ACR} := \frac{1}{|\mathcal{D}_{\texttt{test}}|} \sum_{(x,y) \in \mathcal{D}_{\texttt{test}}} \mathrm{CR}(f, \sigma, x) \cdot \mathbf{1}_{\hat{f}(x)=y}$, where $\mathcal{D}_{\texttt{test}}$ is the test dataset, and CR denotes the certified radius returned from $\mathrm{CERTIFY}(f, \sigma, x)$.

**Training details.** We use the same base classifier used in the prior work [10, 32, 44]: namely, we use LeNet [21] for MNIST, ResNet-110 [16] for CIFAR-10, and ResNet-50 [16] for ImageNet. For a fair comparison, we follow the same training details used in Cohen et al. [10] and Salman et al. [32]. For each model configuration, we consider three different models as varying the noise level $\sigma \in \{0.25, 0.5, 1.0\}$. During inference, we apply randomized smoothing with the same $\sigma$ used in the training. When our regularization is used, we use $m = 2$ and $\eta = 0.5$ unless otherwise specified. More training details are specified in the supplementary material.

**Baseline methods.** We evaluate how consistency regularization would affect the certified robustness when applied to a baseline training method. In our experiments, we consider two baseline methods proposed for training smoothed classifiers to apply our regularization scheme: (a) Gaussian [10]: training with Gaussian augmentation over $\mathcal{N}(0, \sigma^2 I)$; (b) SmoothAdv [32]: adversarial training on a soft approximation of the smoothed classifier. We also consider stability training [23] in (9) and MACER [44] to compare, as other regularization-based approaches.

## 4.2 Results on CIFAR-10

We train CIFAR-10 models for 150 epochs following the training details of SmoothAdv [32]. Whenever possible, we use the pre-trained models officially released by the authors for our evaluation to reproduce the baseline results.[678] For the SmoothAdv models, we report the results for two different

Table 1: Comparison of approximate certified test accuracy (%) on CIFAR-10. Every model is certified with $\sigma$ used for its training. We set our result bold-faced whenever the value improves the baseline. For ACRs, we underline the best model per $\sigma$. For the results in "+ Hyperparameter search", we evaluate the best model among those released by Salman et al. [32] for each $\sigma$.

| $\sigma$ | Models (CIFAR-10) | ACR | 0.00 | 0.25 | 0.50 | 0.75 | 1.00 | 1.25 | 1.50 | 1.75 | 2.00 | 2.25 |
|---|---|---|---|---|---|---|---|---|---|---|---|---|
| 0.25 | Gaussian [10] | 0.424 | 76.6 | 61.2 | 42.2 | 25.1 | 0.0 | 0.0 | 0.0 | 0.0 | 0.0 | 0.0 |
| | + Consistency ($\lambda = 10$) | **0.544** | **77.8** | **68.8** | **57.4** | **43.8** | 0.0 | 0.0 | 0.0 | 0.0 | 0.0 | 0.0 |
| | + Consistency ($\lambda = 20$) | **0.552** | 75.8 | **67.6** | **58.1** | **46.7** | 0.0 | 0.0 | 0.0 | 0.0 | 0.0 | 0.0 |
| | SmoothAdv [32] | 0.544 | 73.4 | 65.6 | 57.0 | 47.5 | 0.0 | 0.0 | 0.0 | 0.0 | 0.0 | 0.0 |
| | + Consistency ($\lambda = 2$) | **0.548** | 72.9 | 65.6 | **57.5** | **48.5** | 0.0 | 0.0 | 0.0 | 0.0 | 0.0 | 0.0 |
| | Stability training [23] | 0.421 | 72.3 | 58.0 | 43.3 | 27.3 | 0.0 | 0.0 | 0.0 | 0.0 | 0.0 | 0.0 |
| | MACER [44] | 0.531 | 79.5 | 69.0 | 55.8 | 40.6 | 0.0 | 0.0 | 0.0 | 0.0 | 0.0 | 0.0 |
| 0.50 | Gaussian [10] | 0.525 | 65.7 | 54.9 | 42.8 | 32.5 | 22.0 | 14.1 | 8.3 | 3.9 | 0.0 | 0.0 |
| | + Consistency ($\lambda = 10$) | **0.720** | 64.3 | **57.5** | **50.6** | **43.2** | **36.2** | **29.5** | **22.8** | **16.1** | 0.0 | 0.0 |
| | SmoothAdv [32] | 0.689 | 64.4 | 57.2 | 49.0 | 40.6 | 33.6 | 27.4 | 21.8 | 14.0 | 0.0 | 0.0 |
| | + Hyperparameter search | 0.717 | 53.1 | 49.2 | 44.9 | 41.0 | 37.2 | 33.2 | 29.1 | 24.0 | 0.0 | 0.0 |
| | + Consistency ($\lambda = 1$) | **0.726** | 52.3 | 48.9 | **45.1** | **41.3** | **37.8** | **33.9** | **29.9** | **25.2** | 0.0 | 0.0 |
| | Stability training [23] | 0.521 | 60.6 | 51.5 | 41.4 | 32.5 | 23.9 | 15.3 | 9.6 | 5.0 | 0.0 | 0.0 |
| | MACER [44] | 0.691 | 64.2 | 57.5 | 49.9 | 42.3 | 34.8 | 27.6 | 20.2 | 12.6 | 0.0 | 0.0 |
| 1.00 | Gaussian [10] | 0.542 | 47.2 | 39.2 | 34.0 | 27.8 | 21.6 | 17.4 | 14.0 | 11.8 | 10.0 | 7.6 |
| | + Consistency ($\lambda = 5$) | **0.734** | **48.1** | **43.9** | **39.3** | **34.7** | **29.9** | **26.1** | **22.1** | **18.8** | **15.4** | **12.2** |
| | + Consistency ($\lambda = 10$) | **0.756** | 46.3 | **42.2** | **38.1** | **34.3** | **30.0** | **26.3** | **22.9** | **19.7** | **16.6** | **13.8** |
| | SmoothAdv [32] | 0.682 | 50.2 | 44.0 | 37.6 | 33.8 | 28.8 | 24.0 | 20.2 | 15.8 | 13.2 | 10.2 |
| | + Hyperparameter search | 0.785 | 45.6 | 41.9 | 38.0 | 34.2 | 30.9 | 27.4 | 24.1 | 20.7 | 17.7 | 14.9 |
| | + Consistency ($\lambda = 1$) | **0.816** | 41.7 | 39.0 | 36.2 | 33.5 | 30.7 | **27.6** | **24.7** | **22.0** | **19.5** | **17.3** |
| | Stability training [23] | 0.526 | 43.5 | 38.9 | 32.8 | 27.0 | 23.1 | 19.1 | 15.4 | 11.3 | 7.8 | 5.7 |
| | MACER [44] | 0.744 | 41.4 | 38.5 | 35.2 | 32.3 | 29.3 | 26.4 | 23.4 | 20.2 | 17.4 | 14.5 |

Table 2: Comparison of approximate certified test accuracy (%) on ImageNet. We set our result bold-faced whenever the value improves the baseline. We use $\eta = 0.1$ instead of $0.5$ when $\sigma = 1.0$.

| $\sigma$ | Models (ImageNet) | ACR | 0.0 | 0.5 | 1.0 | 1.5 | 2.0 | 2.5 | 3.0 | 3.5 |
|---|---|---|---|---|---|---|---|---|---|---|
| 0.50 | Gaussian [10] | 0.733 | 57 | 46 | 37 | 29 | 0 | 0 | 0 | 0 |
| | + Consistency ($\lambda = 5$) | **0.822** | 55 | **50** | **44** | **34** | 0 | 0 | 0 | 0 |
| | SmoothAdv [32] | 0.825 | 54 | 49 | 43 | 37 | 0 | 0 | 0 | 0 |
| 1.00 | Gaussian [10] | 0.875 | 44 | 38 | 33 | 26 | 19 | 15 | 12 | 9 |
| | + Consistency ($\lambda = 5$) | **0.982** | 41 | 37 | 32 | **28** | **24** | **21** | **17** | **14** |
| | SmoothAdv [32] | 1.040 | 40 | 37 | 34 | 30 | 27 | 25 | 20 | 15 |

configurations: (a) for a fixed, pre-defined configuration across $\sigma$, and (b) for the "best" configuration per each $\sigma$, which is heavily examined by Salman et al. [32] over hundreds of models. In case of $\sigma = 0.25$, however, we only report (b) as they show nearly identical results. For (a), we consider a 10-step PGD attack constrained in $\ell_2$ ball of radius $\varepsilon = 1.0$, using $m = 8$ noise samples.[9] In case of stability training [23], we report the best models in terms of ACR across varying $\lambda$ tested: namely, we consider $\lambda \in \{1, 2, 5, 10, 20\}$ for each $\sigma$, and report $\lambda = 2$ for $\sigma = 0.25, 0.5$ and $\lambda = 1$ for $\sigma = 1.0$. The full results can be found in the supplementary material.

The results are presented in Table 1. We also plot certified accuracy over the full range of radii per $\sigma$ in Figure 2. Overall, we observe that our consistency regularization significantly and consistently improves Gaussian and SmoothAdv baselines, both in certified test accuracy and ACR. Specifically, when $\sigma = 0.50$, we found our regularization with $\lambda = 10$ applied on the naïve Gaussian baseline could surpass the best-performing SmoothAdv model reported in "SmoothAdv + Hyperparameter search", in terms of ACR. Furthermore, in case of $\sigma = 1.00$, consistency regularization upon the best SmoothAdv model even further improve the current state-of-the-art baseline by a significant margin, which verifies an orthogonal contribution of our method compared to the prior work. These observations suggest our method works better on more complex tasks, where forcing "confident"

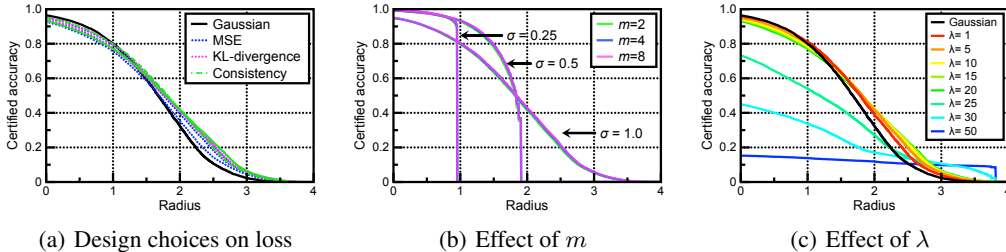

|     (a) Design choices on loss     |     (b) Effect of $m$     |     (c) Effect of $\lambda$     |

Figure 3: Comparison of approximate certified accuracy via randomized smoothing across various types of ablation models. A sharp drop of certified accuracy in the plots exists since there is a hard upper bound that CERTIFY can output for a given $\sigma$ and $n = 100,000$.

prediction (as done in the prior works) might be difficult. We also notice that, despite its similarity with our method, the stability training [23] itself does not improve ACRs even compared to the Gaussian baselines. This is because this training (9) would require $f$ to perform well both in $x$ and $x + \delta$, which is harder to force compared to that of (11) in the context of randomized smoothing.

## 4.3 Results on ImageNet

We also evaluate our regularization scheme on ImageNet classification dataset, to show that our method is scalable on large-scale datasets. We train each model on $\sigma \in \{0.5, 1.0\}$ for 90 epochs. We perform our evaluation on a subsampled test dataset of 500 samples as done by Cohen et al. [10]. As presented in Table 2, we observe that consistency regularization still effectively improves the certified robustness, both in terms of ACR and certified test accuracy, despite its simple and efficient nature of our method. Compared to the best results of SmoothAdv [32], our results achieve a comparable robustness, despite using a single fixed configuration of hyperparameter, namely $\lambda = 5$.

## 4.4 Runtime analysis

With much effectiveness on the certified robustness, consistency regularization also offers a great efficiency in terms of training costs compared to other competitive methods. We compare our method with the baselines in several training statistics, including the number of hyper-parameter (# HP), ACR, memory usage in GPU on peak computation (Mem.), and the total training time (Time). In this experiment, every model is trained on

Table 3: Comparison of training time statistics on CIFAR-10 with $\sigma = 0.50$. All the baselines are trained on their official implementations separately.

| Models | # HP | ACR | Mem. | Time (h) |
|---|---|---|---|---|
| Gaussian | | 0.525 | 2.9G | 4.6 |
| **+ Consistency** | **2** | **0.720** | **2.9G** | **8.7** |
| SmoothAdv | 4 | 0.717 | 3.0G | 23.1 |
| MACER | 4 | 0.691 | 9.4G | 14.1 |

CIFAR-10 using one GPU of NVIDIA TITAN X (Pascal). We use $\sigma = 0.5$ with hyperparameters specified in Section 4.2. In case of SmoothAdv, we choose the best-performing configuration for training. Our method to compare runs upon the Gaussian baseline with $m = 2$ and $\lambda = 10$.

The results in Table 3 show that our regularization indeed costs about twice the Gaussian baseline due to additional sampling, but one can immediately notice that this overhead is far less than others, e.g., compared to adversarial training. Furthermore, our method even achieves better ACR than other methods, which verifies a clear efficiency of consistency regularization compared to the prior work.

## 4.5 Ablation study

We conduct an ablation study for a detailed analysis on our method. Unless otherwise noted, we perform experiments on MNIST in this section. When consistency regularization is used, we assume it is applied upon Gaussian training. We report all the detailed results in the supplementary material.

**Design choices on loss.** We first examine two other popular designs for consistency regularization instead of (7), namely, mean-squared-error [31] and KL-divergence [25] as follow:

$$L^{\mathtt{MSE}} := \lambda \cdot ||F(x + \delta_1) - F(x + \delta_2)||_2^2 \quad \text{and} \quad L^{\mathtt{KL}} := \lambda \cdot \mathbb{E}_\delta[\mathrm{KL}(\hat{F}(x) \,||\, F(x + \delta))], \quad (13)$$

where $\delta_1, \delta_2 \sim \mathcal{N}(0, \sigma^2 I)$. We evaluate certified test accuracy on $\sigma = 1.00$ for these regularization with varying $\lambda \in \{5, 20, 50\}$, and compare the results with $L^{\mathtt{con}}$ with $\lambda \in \{5, 20\}$. The results are

Table 4: Comparison of ACR on CIFAR-10 with $\sigma = 0.5$ for different architectures. Bold indicates the best ACR value per architecture. For "+ Best HP (ResNet-110)", we use the hyperparameters those are optimized for ResNet-110 by Salman et al. [32], as used in Table 1.

| Architecture | CIFAR-10 ($\sigma = 0.5$) | ACR | 0.00 | 0.25 | 0.50 | 0.75 | 1.00 | 1.25 | 1.50 | 1.75 |
|---|---|---|---|---|---|---|---|---|---|---|
| ResNet-20 | Gaussian [10] | 0.524 | 67.0 | 55.4 | 42.8 | 31.4 | 22.0 | 13.9 | 8.1 | 3.8 |
| | **+ Consistency** ($\lambda = 10$) | **0.686** | 60.5 | 54.4 | **47.7** | **40.9** | **34.4** | **28.0** | **22.2** | **16.5** |
| | SmoothAdv [32] | 0.692 | 63.0 | 56.5 | 48.9 | 41.8 | 34.9 | 28.1 | 21.3 | 14.6 |
| | + Best HP (ResNet-110) | 0.682 | 50.8 | 47.0 | 43.1 | 39.0 | 35.1 | 31.6 | 27.2 | 22.7 |
| | Stability training [23] | 0.499 | 60.2 | 50.1 | 40.5 | 31.0 | 22.1 | 14.9 | 8.2 | 3.7 |
| | MACER [44] | 0.661 | 63.0 | 55.7 | 48.2 | 40.5 | 32.6 | 25.5 | 18.7 | 11.9 |
| DenseNet-40 | Gaussian [10] | 0.494 | 65.0 | 53.7 | 41.2 | 29.6 | 19.6 | 12.4 | 6.9 | 3.1 |
| | **+ Consistency** ($\lambda = 10$) | **0.661** | 59.1 | 52.7 | **46.1** | **39.3** | **32.8** | **27.0** | **21.1** | **15.6** |
| | SmoothAdv [32] | 0.671 | 61.6 | 55.3 | 48.0 | 40.3 | 33.2 | 26.4 | 20.4 | 14.3 |
| | + Best HP (ResNet-110) | 0.659 | 49.4 | 45.8 | 41.5 | 37.6 | 33.9 | 30.2 | 26.5 | 22.1 |
| | Stability training [23] | 0.497 | 56.5 | 47.9 | 38.8 | 30.7 | 23.0 | 16.5 | 9.9 | 4.9 |
| | MACER [44] | 0.641 | 62.0 | 54.5 | 46.7 | 39.1 | 31.8 | 24.8 | 17.8 | 11.4 |

presented in Figure 3(a). In general, we observe that both regularizers, namely $L^{\mathtt{MSE}}$ and $L^{\mathtt{KL}}$, are also capable to improve the certified robustness, but they could not achieve a better ACR than $L^{\mathtt{con}}$ even with a moderately large $\lambda$. Considering that $L^{\mathtt{KL}}$ is equivalent to $L^{\mathtt{con}}$ when $\eta = 0$ in (7), this observation indicates the importance of regularizing the *entropy* of the mean prediction. Indeed, we empirically observe that both $L^{\mathtt{MSE}}$ and $L^{\mathtt{KL}}$ often lead the predictions to be too close to the uniform when $\lambda$ is large, which may harm the discriminative performance of the base classifier.

**Effect of $m$.** As mentioned in Section 3, the computational costs for our regularization scheme highly depends on the number of noise samples used, namely $m > 1$. Nevertheless, we observe that our regularization is fairly robust on the choice of $m$, so that $m = 2$ usually leads to good enough performance. In Figure 3(b), we compare the certified robustness of models trained with our regularization with varying $m \in \{2, 4, 8\}$. For each $m$, we present three different models under various $\sigma \in \{0.25, 0.5, 1.0\}$. The results show that models using $m = 2$ perform nearly identically to others, while one could observe slight improvements for larger $m$. In practice, this observation reduces much of the hyperparameter complexity in our method: by simply letting $m$ to be small, e.g., $m = 2$, while fixing $\eta = 0.5$, $\lambda$ becomes the only crucial hyperparameter.

**Effect of $\lambda$.** We also investigate the effect of having different $\lambda$ in Figure 3(c). As expected, we observe a clear trade-off between accuracy and robustness of the corresponding smoothed classifier by controlling $\lambda$. Furthermore, for a sufficiently large $\lambda$, e.g., $\lambda = 50$ in Figure 3(c), a classifier is often trained to return maximal certifiable radius for any input when smoothed, even the accuracy falls into chance-level. Nevertheless, this would be a desirable property for a trade-off term between accuracy and robustness, which has not been explored much for smoothed classifiers.

**Different architectures.** In our experiments, we follow the prior works [10, 32, 44] to choose network architectures for a fair comparison, e.g., we use ResNet-110 [16] for CIFAR-10. To explore the effect of different architectures, we further test our method with ResNet-20 [16] and DenseNet-40 [18] on CIFAR-10, as summarized in Table 4.5. We assume $\sigma = 0.5$ for this experiment, and use the same hyperparameters specified in Section 4.2. Overall, our method also consistently outperforms other baselines except SmoothAdv on these architectures. In particular, it is remarkable that our choice of hyperparameter, namely $\lambda = 10$, transfers well to other architectures, compared to SmoothAdv: the best working hyperparameters for ResNet-110 could not further improve SmoothAdv from the baseline configuration, i.e., it may require a further optimization to perform better.

## 5 Conclusion

In this paper, we show *consistency regularization* can play a key role in certifiable robustness of smoothed classifiers. We think our work would emphasize the importance of *noise-consistent* inference in deep neural networks, one of under-explored topics despite its desirable property. We also expect our work can be a useful guideline when other researchers will study the noise-consistency in other problems in the future. Many questions are related: how can we design a noise-invariant neural network, or for which family of noise this would be allowed, just to name a few.

## Broader Impact

The potential risk of adversarial attacks has left many practitioners hesitant to apply the latest developments in deep learning into their systems. Adversarial robustness of deep neural networks is one of the most important research problems toward *AI safety* [1], with much impact on various applications especially for security-concerned systems: e.g., medical diagnosis [9], speech recognition [29], and autonomous driving [43]. Our research could be beneficial for those who design such systems, thanks to the certifiable guarantees on adversarial robustness that *randomized smoothing* can provide. Especially, we expect the simplicity of our method would encourage many practitioners to incorporate randomized smoothing into their systems along with our work. A practical success of systems equipped with a sufficient amount of certified robustness would be fatal for those who maliciously attempt to break down the system via adversarial attacks.

This statement, however, presumes that many of the practical issues on the current randomized smoothing technique would be resolved in future research. For example, (a) randomized smoothing requires exponentially many inferences for a single reliable inference, and (b) there is still a gap between theoretical guarantee [13, 42] and practice [40, 4] on robustness that randomized smoothing currently gives: consequently, current randomized smoothing can be easily misused in practical systems, and a failure of such systems may implicitly lead practitioners to have a biased, false sense of security. We believe our research is a step toward reducing this practical gap to deploy randomized smoothing into the real-world.

## Acknowledgments and Disclosure of Funding

This work was conducted by Center for Applied Research in Artificial Intelligence(CARAI) grant funded by Defense Acquisition Program Administration(DAPA) and Agency for Defense Development(ADD) (UD190031RD).

## Footnotes

[1]For the interested readers, we present a detailed overview of prior works in the supplementary material.

[2]We remark the regularization term in (12) requires $m > 1$ to work, as the KL term in (7) would vanish when $m = 1$: with only a single sample, say $\delta_1$, $F(x + \delta_1)$ would be the best estimation of $\hat{F}(x)$.

[3]All the results on MNIST are provided in the supplementary material.

[4]Code is available at https://github.com/jh-jeong/smoothing-consistency.

[5]https://github.com/locuslab/smoothing

[6]https://github.com/Hadisalman/smoothing-adversarial

[7]https://github.com/RuntianZ/macer

[8]In case of the pre-trained MACER models, we observe a slight discrepancy between our evaluation and those reported in Zhai et al. [44]. We have verified that this is due to a sampling bias: we found [44] used 500 contiguous subsamples by default in the official code, while our evaluation uses the full CIFAR-10 test set.

[9]The detailed configurations for (b), the best models, are specified in the supplementary material.

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
