[Supplementary Material]

# Supplementary Material:

## Consistency Regularization for Certified Robustness of Smoothed Classifiers

## A   Details on experimental setups

### A.1   Training details

We train every model via stochastic gradient descent (SGD) with Nesterov momentum of weight 0.9 without dampening. We set a weight decay of $10^{-4}$ for all the models. We use different training schedules for each dataset: (a) MNIST: The initial learning rate is set to 0.01; We train a model for 90 epochs with mini-batch size 256, and the learning rate is decayed by 0.1 at 30-th and 60-th epoch, (b) CIFAR-10: The initial learning rate is set to 0.1; We train a model for 150 epochs with mini-batch size 256, and the learning rate is decayed by 0.1 at 50-th and 100-th epoch, and (c) ImageNet: The initial learning rate is set to 0.1; We train a model for 90 epochs with mini-batch size 200, and the learning rate is decayed by 0.1 at 30-th and 60-th epoch. When SmoothAdv is used, we adopt the *warm-up* strategy on attack radius $\varepsilon$ [8], i.e., $\varepsilon$ is initially set to zero, and linearly increased during the first 10 epochs to a pre-defined hyperparameter.

### A.2   Datasets

**MNIST** dataset [3] consists 70,000 gray-scale hand-written digit images of size $28 \times 28$, 60,000 for training and 10,000 for testing. Each of the images is labeled from 0 to 9, i.e., there are 10 classes. When training on MNIST, we do not perform any pre-processing except for normalizing the range of each pixel from 0-255 to 0-1. The full dataset can be downloaded at `http://yann.lecun.com/exdb/mnist/`.

**CIFAR-10** dataset [2] consist of 60,000 RGB images of size $32 \times 32$ pixels, 50,000 for training and 10,000 for testing. Each of the images is labeled to one of 10 classes, and the number of data per class is set evenly, i.e., 6,000 images per each class. We follow the same data-augmentation scheme used in Cohen et al. [1], Salman et al. [8] for a fair comparison, namely, we use random horizontal flip and random translation up to 4 pixels. We also normalize the images in pixel-wise by the mean and the standard deviation calculated from the training set. Here, an important practical point is that this normalization is done *after* a noise is added to input when regarding randomized smoothing, following Cohen et al. [1]. This is to ensure that noise is given to the original image coordinates. In practical implementations, this can be done by placing the normalization as the first layer of base classifiers, instead of as a pre-processing step. The full dataset can be downloaded at `https://www.cs.toronto.edu/~kriz/cifar.html`

**ImageNet** classification dataset [7] consists of 1.2 million training images and 50,000 validation images, which are labeled by one of 1,000 classes. For data-augmentation, we perform $224 \times 224$ random cropping with random resizing and horizontal flipping to the training images. At test time, on the other hand, $224 \times 224$ center cropping is performed after re-scaling the images into $256 \times 256$. This pre-processing scheme is also used in Cohen et al. [1], Salman et al. [8] as well. Similar to CIFAR-10, all the images are normalized *after* adding a noise in pixel-wise by the pre-computed mean and standard deviation. A link for downloading the full dataset can be found in `http://image-net.org/download`.

Table 1: Detailed specification of hyperparameters used in the best-performing SmoothAdv models.

| Dataset | $\sigma$ | Method | # steps | $\varepsilon$ | $m$ |
|---------|------|--------|---------|------|-----|
| CIFAR-10 | 0.25 | PGD | 10 | 255 | 4 |
|  | 0.50 | PGD | 10 | 512 | 2 |
|  | 1.00 | PGD | 10 | 512 | 2 |
| ImageNet | 0.50 | PGD | 1 | 255 | 1 |
|  | 1.00 | PGD | 1 | 512 | 1 |

Table 2: Comparison of approximate certified test accuracy on MNIST dataset. For each model, training and certification are done with the same smoothing factor specified in $\sigma$. Each of the values indicates the fraction of test samples those have $\ell_2$ certified radius larger than the threshold specified at the top row. We set our result bold-faced whenever the value improves the baseline. For ACR, we underlined the best-performing model per each $\sigma$.

| $\sigma$ | Models (MNIST) | ACR | 0.00 | 0.25 | 0.50 | 0.75 | 1.00 | 1.25 | 1.50 | 1.75 | 2.00 | 2.25 | 2.50 |
|---|---|---|---|---|---|---|---|---|---|---|---|---|---|
| | Gaussian [1] | 0.911 | 99.2 | 98.5 | 96.7 | 93.3 | 0.0 | 0.0 | 0.0 | 0.0 | 0.0 | 0.0 | 0.0 |
| | + Consistency ($\lambda = 20$) | **0.930** | **99.5** | **99.0** | **98.0** | **96.4** | 0.0 | 0.0 | 0.0 | 0.0 | 0.0 | 0.0 | 0.0 |
| 0.25 | SmoothAdv [8] | 0.932 | 99.4 | 99.0 | 98.2 | 96.8 | 0.0 | 0.0 | 0.0 | 0.0 | 0.0 | 0.0 | 0.0 |
| | + Consistency ($\lambda = 5$) | 0.932 | 99.2 | 98.8 | 98.0 | 96.8 | 0.0 | 0.0 | 0.0 | 0.0 | 0.0 | 0.0 | 0.0 |
| | Stability training [4] | 0.915 | 99.3 | 98.6 | 97.1 | 93.8 | 0.0 | 0.0 | 0.0 | 0.0 | 0.0 | 0.0 | 0.0 |
| | MACER [9] | 0.920 | 99.3 | 98.7 | 97.5 | 94.8 | 0.0 | 0.0 | 0.0 | 0.0 | 0.0 | 0.0 | 0.0 |
| | Gaussian [1] | 1.553 | 99.2 | 98.3 | 96.8 | 94.3 | 89.7 | 81.9 | 67.3 | 43.6 | 0.0 | 0.0 | 0.0 |
| | + Consistency ($\lambda = 20$) | **1.665** | 98.9 | 98.3 | **97.4** | **95.6** | **93.0** | **88.1** | **79.5** | **62.9** | 0.0 | 0.0 | 0.0 |
| 0.50 | SmoothAdv [8] | 1.687 | 99.0 | 98.3 | 97.3 | 95.8 | 93.2 | 88.5 | 81.1 | 67.5 | 0.0 | 0.0 | 0.0 |
| | + Consistency ($\lambda = 5$) | **1.690** | 98.2 | 97.5 | 96.5 | 94.7 | 91.7 | 87.7 | **81.5** | **71.2** | 0.0 | 0.0 | 0.0 |
| | Stability training [4] | 1.570 | 99.2 | 98.5 | 97.1 | 94.8 | 90.7 | 83.2 | 69.2 | 45.4 | 0.0 | 0.0 | 0.0 |
| | MACER [9] | 1.594 | 98.5 | 97.5 | 96.2 | 93.7 | 90.0 | 83.7 | 72.2 | 54.0 | 0.0 | 0.0 | 0.0 |
| | Gaussian [1] | 1.620 | 96.4 | 94.4 | 91.4 | 87.0 | 79.9 | 71.0 | 59.6 | 46.2 | 32.6 | 19.7 | 10.8 |
| | + Consistency ($\lambda = 20$) | **1.750** | 92.5 | 89.8 | 86.5 | 81.7 | 76.3 | 69.6 | **61.4** | **52.9** | **44.2** | **35.9** | **26.8** |
| 1.00 | SmoothAdv [8] | 1.779 | 95.8 | 93.9 | 90.6 | 86.5 | 80.8 | 73.7 | 64.6 | 53.9 | 43.3 | 32.8 | 22.2 |
| | + Consistency ($\lambda = 5$) | **1.811** | 92.9 | 90.4 | 87.0 | 82.5 | 77.0 | 70.2 | 62.7 | 53.7 | **45.5** | **37.7** | **29.4** |
| | Stability training [4] | 1.634 | 96.5 | 94.6 | 91.7 | 87.4 | 80.6 | 72.0 | 60.5 | 46.8 | 33.1 | 20.0 | 11.2 |
| | MACER [9] | 1.570 | 92.0 | 88.5 | 84.0 | 78.1 | 71.5 | 63.8 | 55.3 | 46.3 | 36.5 | 26.2 | 16.3 |

(a) $\sigma = 0.25$     (b) $\sigma = 0.50$     (c) $\sigma = 1.00$

Figure 1: Comparison of approximate certified accuracy via randomized smoothing for various training methods on MNIST. A sharp drop of certified accuracy in the plots exists since there is a hard upper bound that CERTIFY can output for a given $\sigma$ and $n = 100,000$.

### A.3 Detailed configurations of SmoothAdv models

In Table 1, we specify the exact configurations used in our evaluation for the best-performing SmoothAdv models. These configurations have originally explored by Salman et al. [8] via a grid search over 4 hyperparameters: namely, (a) attack method (Method): PGD [5] or DDN [6], (b) the number of steps (# steps), (c) the maximum allowed $\ell_2$ perturbation on the input ($\varepsilon$), and (d) the number of noise samples ($m$). We choose one pre-trained model per $\sigma$ for CIFAR-10 and ImageNet, among those officially released and classified as the best-performing models by Salman et al. [8]. The link to download all the pre-trained models can be found in https://github.com/Hadisalman/smoothing-adversarial.

## B Results on MNIST

We train every MNIST model for 90 epochs. We consider a fixed configuration of hyperparameters when SmoothAdv is used in MNIST: specifically, we perform a 10-step projected gradient descent (PGD) attack constrained in $\ell_2$ ball of radius $\varepsilon = 1.0$ for each input, while the objective is approximated with $m = 4$ noise samples. For the MACER models, on the other hand, we generally follow the hyperparameters specified in the original paper [9]: we set $m = 16$, $\lambda = 16.0$, $\gamma = 8.0$ and

$\beta = 16.0.$[1] In $\sigma = 1.0$, however, we had to reduce $\lambda$ to 6 for a successful training. Nevertheless, we have verified that the ACRs computed from the reproduced models are comparable to those reported in the original paper. We use $\lambda = 2$ when stability training [4] is applied in this section.

We report the results in Table 2 and Figure 1. Overall, we observe that our consistency regularization stably improve Gaussian and SmoothAdv baselines in ACR, except when applied to SmoothAdv on $\sigma = 0.25$. This corner-case is possibly due to that the model is already achieve to the best capacity via SmoothAdv, regarding that MNIST on $\sigma = 0.25$ is relatively a trivial task. For the rest non-trivial cases, nevertheless, our regularization shows a remarkable effectiveness in two aspects: (a) applying our consistency regularization on Gaussian, the simplest baseline, dramatically improves the certified test accuracy and ACR even outperforming the recently proposed MACER by a large margin, and (b) when applied to SmoothAdv, our method could further improve ACR. In particular, one could observe that our regularization significantly improves the certified accuracy especially at large radii, where a classifier should attain a high value of $p^{(1)}$ (5), i.e., a consistent prediction is required.

## C  Variance of results over multiple runs

In our experiments, we compare single-run results following other baselines considered in this paper [1, 8, 4, 9]. In Table 3, we report the mean and standard deviation of ACRs across 5 seeds for the MNIST results reported in Table 2. In general, we observe ACR of a given training method is fairly robust to network initialization.

Table 3: Comparison of ACR for various training methods on MNIST. The reported values are the mean and standard deviation across 5 seeds. We set our result bold-faced whenever the value improves the baseline, and the underlined are best-performing model per $\sigma$.

| ACR (MNIST) | $\sigma = 0.25$ | $\sigma = 0.50$ | $\sigma = 1.00$ |
|---|---|---|---|
| Gaussian [1] | $0.9108_{\pm 0.0003}$ | $1.5581_{\pm 0.0016}$ | $1.6184_{\pm 0.0021}$ |
| **+ Consistency** | $\mathbf{0.9300}_{\pm \mathbf{0.0004}}$ | $\mathbf{1.6653}_{\pm \mathbf{0.0007}}$ | $\mathbf{1.7486}_{\pm \mathbf{0.0025}}$ |
| SmoothAdv [8] | $\underline{0.9322}_{\pm 0.0005}$ | $1.6872_{\pm 0.0007}$ | $1.7786_{\pm 0.0017}$ |
| **+ Consistency** | $0.9318_{\pm 0.0002}$ | $\underline{\mathbf{1.6901}}_{\pm \mathbf{0.0009}}$ | $\underline{\mathbf{1.8082}}_{\pm \mathbf{0.0026}}$ |
| Stability [4] | $0.9152_{\pm 0.0007}$ | $1.5719_{\pm 0.0028}$ | $1.6341_{\pm 0.0018}$ |
| MACER [9] | $0.9201_{\pm 0.0006}$ | $1.5899_{\pm 0.0069}$ | $1.5950_{\pm 0.0051}$ |

## D  Detailed results in ablation study

We report the detailed results for the experiments performed in ablation study (see Section 4.6 in the main text). Table 4, 5, and 6 are corresponded to Figure 4(a), 4(b), and 4(c) in the main text, respectively.

Table 4: Comparison of approximate certified test accuracy (%) on MNIST, for varing loss functions and $\lambda$. We set our result bold-faced whenever the value improves the baseline. For ACR, we underlined the best-performing model.

| Model | $\lambda$ | ACR | 0.00 | 0.25 | 0.50 | 0.75 | 1.00 | 1.25 | 1.50 | 1.75 | 2.00 | 2.25 | 2.50 |
|---|---|---|---|---|---|---|---|---|---|---|---|---|---|
| Gaussian | 0 | 1.620 | 96.4 | 94.4 | 91.4 | 87.0 | 79.9 | 71.0 | 59.6 | 46.2 | 32.6 | 19.7 | 10.8 |
| MSE | 20 | **1.677** | 93.6 | 91.0 | 87.5 | 83.0 | 77.1 | 69.9 | **60.8** | **50.3** | **39.5** | **28.6** | **18.4** |
| | 50 | 1.603 | 92.5 | 90.0 | 86.1 | 81.3 | 75.5 | 67.7 | 58.6 | **47.4** | **35.7** | **24.1** | **14.5** |
| | 100 | 1.570 | 91.5 | 88.8 | 84.7 | 80.0 | 73.8 | 65.9 | 57.0 | **46.5** | **34.6** | **23.5** | **14.3** |
| KL-divergence | 20 | **1.713** | 94.0 | 91.7 | 88.2 | 83.5 | 77.7 | 70.5 | **61.5** | **51.4** | **41.2** | **31.1** | **21.4** |
| | 50 | **1.707** | 93.4 | 90.7 | 87.1 | 82.3 | 76.8 | 69.4 | **60.6** | **50.9** | **41.3** | **31.8** | **22.6** |
| | 100 | **1.683** | 92.7 | 89.9 | 85.9 | 81.3 | 75.4 | 68.0 | 59.5 | **49.9** | **40.4** | **31.2** | **22.6** |
| Cross-entropy | 20 | $\underline{\mathbf{1.750}}$ | 92.5 | 89.8 | 86.5 | 81.7 | 76.3 | 69.6 | **61.4** | **52.9** | **44.2** | **35.9** | **26.8** |

Table 5: Comparison of approximate certified test accuracy on MNIST for varying $m \in \{2, 4, 8\}$. For each model, training and certification are done with the same smoothing factor specified in $\sigma$.

| $\sigma$ | $m$ | ACR | 0.00 | 0.25 | 0.50 | 0.75 | 1.00 | 1.25 | 1.50 | 1.75 | 2.00 | 2.25 | 2.50 |
|---|---|---|---|---|---|---|---|---|---|---|---|---|---|
| | 2 | 0.930 | 99.5 | 99.0 | 98.0 | 96.4 | 0.0 | 0.0 | 0.0 | 0.0 | 0.0 | 0.0 | 0.0 |
| 0.25 | 4 | 0.931 | 99.5 | 99.0 | 98.2 | 96.5 | 0.0 | 0.0 | 0.0 | 0.0 | 0.0 | 0.0 | 0.0 |
| | 8 | 0.931 | 99.4 | 98.9 | 98.2 | 96.5 | 0.0 | 0.0 | 0.0 | 0.0 | 0.0 | 0.0 | 0.0 |
| | 2 | 1.665 | 98.9 | 98.3 | 97.4 | 95.6 | 93.0 | 88.1 | 79.5 | 62.9 | 0.0 | 0.0 | 0.0 |
| 0.50 | 4 | 1.672 | 99.0 | 98.3 | 97.4 | 95.6 | 93.2 | 88.7 | 80.0 | 63.7 | 0.0 | 0.0 | 0.0 |
| | 8 | 1.670 | 98.9 | 98.4 | 97.4 | 95.8 | 93.2 | 88.7 | 79.9 | 63.6 | 0.0 | 0.0 | 0.0 |
| | 2 | 1.750 | 92.5 | 89.8 | 86.5 | 81.7 | 76.3 | 69.6 | 61.4 | 52.9 | 44.2 | 35.9 | 26.8 |
| 1.00 | 4 | 1.762 | 92.2 | 89.4 | 86.3 | 81.5 | 76.0 | 69.3 | 61.2 | 52.6 | 44.2 | 36.9 | 28.1 |
| | 8 | 1.771 | 91.9 | 89.0 | 85.9 | 81.5 | 76.1 | 69.2 | 61.4 | 52.8 | 44.9 | 37.2 | 29.0 |

Table 6: Comparison of approximate certified test accuracy on MNIST for varying $\lambda$. We set our result bold-faced whenever the value improves the baseline ($\lambda = 0.0$). For ACR, we underlined the best-performing model.

| $\lambda$ | ACR | 0.00 | 0.25 | 0.50 | 0.75 | 1.00 | 1.25 | 1.50 | 1.75 | 2.00 | 2.25 | 2.50 |
|---|---|---|---|---|---|---|---|---|---|---|---|---|
| 0.0 | 1.620 | 96.4 | 94.4 | 91.4 | 87.0 | 79.9 | 71.0 | 59.6 | 46.2 | 32.6 | 19.7 | 10.8 |
| 1.0 | **1.707** | 96.0 | 94.1 | 91.3 | 86.9 | **81.0** | **73.0** | **63.1** | **51.2** | **38.7** | **26.8** | **15.5** |
| 5.0 | **1.741** | 95.0 | 92.9 | 89.8 | 85.4 | 79.5 | **72.6** | **63.3** | **52.8** | **41.8** | **31.0** | **20.6** |
| 10.0 | **1.743** | 94.2 | 92.0 | 88.8 | 84.3 | 78.7 | **71.7** | **62.9** | **52.8** | **42.7** | **32.3** | **22.7** |
| 15.0 | **1.749** | 93.7 | 91.2 | 87.8 | 83.2 | 77.6 | 70.6 | **62.2** | **52.8** | **43.7** | **34.1** | **24.7** |
| 20.0 | **1.750** | 92.5 | 89.8 | 86.5 | 81.7 | 76.3 | 69.6 | **61.4** | **52.9** | **44.2** | **35.9** | **26.8** |
| 25.0 | **1.737** | 90.5 | 88.0 | 84.3 | 79.7 | 74.4 | 67.6 | **60.6** | **52.6** | **44.8** | **37.3** | **28.2** |
| 30.0 | **1.713** | 87.4 | 84.6 | 81.0 | 76.5 | 71.4 | 65.5 | 59.1 | **52.4** | **45.6** | **38.3** | **29.9** |
| 35.0 | **1.654** | 81.6 | 78.8 | 75.7 | 71.3 | 66.7 | 62.0 | 56.6 | **51.4** | **45.3** | **39.0** | **31.4** |
| 40.0 | 0.432 | 11.3 | 11.3 | 11.3 | 11.3 | 11.3 | 11.3 | 11.3 | 11.3 | 11.3 | 11.3 | **11.3** |

Table 7: Comparison of our method to stability training [4] on CIFAR-10 dataset. Each of the values indicates the fraction of test samples those have $\ell_2$ certified radius larger than the threshold specified at the top row. We set our result bold-faced whenever the value improves the baseline.

| $\sigma$ | Models (CIFAR-10) | ACR | 0.00 | 0.25 | 0.50 | 0.75 | 1.00 | 1.25 | 1.50 | 1.75 | 2.00 | 2.25 |
|---|---|---|---|---|---|---|---|---|---|---|---|---|
| | Gaussian [1] | 0.424 | 76.6 | 61.2 | 42.2 | 25.1 | 0.0 | 0.0 | 0.0 | 0.0 | 0.0 | 0.0 |
| | + Consistency ($\lambda = 20$) | **0.551** | **75.9** | **67.5** | **57.6** | **46.9** | 0.0 | 0.0 | 0.0 | 0.0 | 0.0 | 0.0 |
| 0.25 | Stability [4] ($\lambda = 1$) | 0.408 | 71.6 | 57.8 | 40.7 | 27.0 | 0.0 | 0.0 | 0.0 | 0.0 | 0.0 | 0.0 |
| | Stability [4] ($\lambda = 2$) | 0.421 | 72.3 | 58.0 | 43.3 | 27.3 | 0.0 | 0.0 | 0.0 | 0.0 | 0.0 | 0.0 |
| | Stability [4] ($\lambda = 5, 10, 20$) | 0.102 | 10.7 | 10.7 | 10.7 | 10.7 | 0.0 | 0.0 | 0.0 | 0.0 | 0.0 | 0.0 |
| | Gaussian [1] | 0.525 | 65.7 | 54.9 | 42.8 | 32.5 | 22.0 | 14.1 | 8.3 | 3.9 | 0.0 | 0.0 |
| | + Consistency ($\lambda = 20$) | **0.737** | 59.0 | 53.9 | **48.7** | **43.4** | **37.8** | **32.7** | **27.0** | **21.0** | 0.0 | 0.0 |
| 0.50 | Stability [4] ($\lambda = 1$) | 0.496 | 61.1 | 51.5 | 40.9 | 29.8 | 21.1 | 14.0 | 8.3 | 3.6 | 0.0 | 0.0 |
| | Stability [4] ($\lambda = 2$) | 0.521 | 60.6 | 51.5 | 41.4 | 32.5 | 23.9 | 15.3 | 9.6 | 5.0 | 0.0 | 0.0 |
| | Stability [4] ($\lambda = 5, 10, 20$) | 0.206 | 10.8 | 10.8 | 10.8 | 10.8 | 10.8 | 10.8 | 10.8 | 10.8 | 0.0 | 0.0 |
| | Gaussian [1] | 0.542 | 47.2 | 39.2 | 34.0 | 27.8 | 21.6 | 17.4 | 14.0 | 11.8 | 10.0 | 7.6 |
| | + Consistency ($\lambda = 20$) | **0.742** | 43.8 | **40.1** | **36.0** | **32.3** | **28.4** | **25.0** | **21.7** | **18.9** | **16.3** | **13.8** |
| 1.00 | Stability [4] ($\lambda = 1$) | 0.526 | 43.5 | 38.9 | 32.8 | 27.0 | 23.1 | 19.1 | 15.4 | 11.3 | 7.8 | 5.7 |
| | Stability [4] ($\lambda = 2$) | 0.414 | 17.0 | 16.3 | 15.4 | 14.6 | 13.7 | 12.6 | 12.1 | 11.2 | 10.3 | 9.8 |
| | Stability [4] ($\lambda = 5, 10, 20$) | 0.381 | 10.0 | 10.0 | 10.0 | 10.0 | 10.0 | 10.0 | 10.0 | 10.0 | 10.0 | 10.0 |

# E  Overview on prior works

For completeness, we present a brief introduction to the prior works mainly considered in our experiments. We use the notations defined in Section 2 of the main text throughout this section.

## E.1  SmoothAdv

Recall that a smoothed classifier $\hat{f}$ is defined from a hard classifier $f : \mathbb{R}^d \to \mathcal{Y}$, namely:

$$\hat{f}(x) := \arg\max_{k \in \mathcal{Y}} \mathbb{P}_{\delta \sim \mathcal{N}(0, \sigma^2 I)} \left( f(x + \delta) = k \right). \tag{1}$$

Here, *SmoothAdv* [8] attempts to perform adversarial training [5] directly on $\hat{f}$:

$$\min_{\hat{f}} \max_{||x'-x||_2 \leq \epsilon} \mathcal{L}(\hat{f}; x', y), \tag{2}$$

where $\mathcal{L}$ denotes the standard cross-entropy loss. As mentioned in the main text, however, $\hat{f}$ is practically a non-differentiable object when (1) is approximated via Monte Carlo sampling, making it difficult to optimize the inner maximization of (2). To bypass this, Salman et al. [8] propose to attack the *soft-smoothed* classifier $\hat{F} := \mathbb{E}_\delta[F_y(x + \delta)]$ instead of $\hat{f}$, as $\hat{F} : \mathbb{R}^d \to \Delta^{K-1}$ is rather differentiable. Namely, SmoothAdv finds an adversarial example via solving the following:

$$\hat{x} = \arg\max_{||x'-x||_2 \leq \epsilon} \mathcal{L}(\hat{F}; x', y) = \arg\max_{||x'-x||_2 \leq \epsilon} \left( -\log \mathbb{E}_\delta \left[ F_y(x' + \delta) \right] \right). \tag{3}$$

In practice, the expectation in this objective (3) is approximated via Monte Carlo integration with $m$ samples of $\delta$, namely $\delta_1, \cdots, \delta_m \sim \mathcal{N}(0, \sigma^2 I)$:

$$\hat{x} = \arg\max_{||x'-x||_2 \leq \epsilon} \left( -\log \left( \frac{1}{m} \sum_i F_y(x' + \delta_i) \right) \right). \tag{4}$$

To optimize the outer minimization objective in (2), on the other hand, SmoothAdv simply minimize the averaged loss over $(\hat{x} + \delta_1, y), \cdots, (\hat{x} + \delta_m, y)$, i.e., $\min_F \frac{1}{m} \sum_i \mathcal{L}(F; \hat{x} + \delta_i, y)$. Notice that the noise samples $\delta_1, \cdots, \delta_m$ are re-used for the outer minimization as well.

## E.2  MACER

On the other hand, MACER [9] attempts to improve robustness of $\hat{f}$ via directly maximizing the certified lower bound over $\ell_2$-adversarial perturbation [1] for $(x, y) \in \mathcal{D}$:

$$\min_{\hat{f}(x') \neq y} ||x' - x||_2 \geq \frac{\sigma}{2} \left( \Phi^{-1}(p^{(1)}) - \Phi^{-1}(p^{(2)}) \right), \tag{5}$$

where $p^{(1)} := \mathbb{P}(f(x + \delta) = \hat{f}(x))$ and $p^{(2)} := \max_{c \neq \hat{f}(x)} \mathbb{P}(f(x + \delta) = c)$, as defined in Section 2 in the main text. Again, directly maximizing (5) is difficult due to the non-differentiability of $\hat{f}$, thereby MACER instead maximizes the certified radius of $\hat{F}$, in a similar manner to SmoothAdv [8]:

$$\text{CR}(\hat{F}; x, y) := \frac{\sigma}{2} \left( \Phi^{-1}(\mathbb{E}_\delta[F_y(x + \delta)]) - \Phi^{-1}(\max_{c \neq y} \mathbb{E}_\delta[F_c(x + \delta)]) \right). \tag{6}$$

Motivated from the 0-1 robust classification loss (7), Zhai et al. [9] propose a robust training objective for maximizing $\text{CR}(\hat{F}; x, y)$ along with the standard cross-entropy loss $\mathcal{L}$ on $\hat{F}$ as a surrogate loss for the natural error term:

$$L_\varepsilon(f) := \mathbb{E}_{(x,y) \in \mathcal{D}} \left[ 1 - \mathbf{1}_{\text{CR}(\hat{f}; x, y) \geq \varepsilon} \right] = \underbrace{\mathbb{E} \left[ \mathbf{1}_{\hat{f}(x) \neq y} \right]}_{\text{natural error}} + \underbrace{\mathbb{E} \left[ \mathbf{1}_{\hat{f}(x) = y, \, \text{CR}(\hat{f}; x, y) < \varepsilon} \right]}_{\text{robust error}} \tag{7}$$

$$L_{\texttt{MACER}}(F; x, y) := \underbrace{\mathcal{L}(\hat{F}(x), y)}_{\text{natural error}} + \lambda \cdot \underbrace{\frac{\sigma}{2} \max\{\gamma - \text{CR}(\hat{F}; x, y), 0\} \cdot \mathbf{1}_{\hat{F}(x) = y}}_{\text{robust error}}, \tag{8}$$

where $\gamma, \lambda$ are hyperparameters. Here, notice that (8) uses the hinge loss to maximize $\text{CR}(\hat{F}; x, y)$, only for the samples that $\hat{F}(x)$ is correctly classified to $y$. In addition, MACER uses an inverse temperature $\beta > 1$ to calibrate $\hat{F}$ as another hyperparameter, mainly for reducing the practical gap between $\hat{F}$ and $\hat{f}$.

## Footnotes

[1]We refer the readers to Zhai et al. [9] for the details on each hyperparemeter.