[Reviews · NeurIPS 2020]

Review 1

Summary and Contributions: This paper studied the problem of training of Gaussian smoothed robust classifiers, which was first proposed in Cohen et al. 2019. Following recent work by Zhai et al. 2020, this paper proposed a consistency regularization method to improve the training. Specifically, the authors follows the natural-robust error decomposition by Zhai et al. 2020, and used a different surrogate loss for the certified robust error. The paper is very well-written.

Strengths: Simplicity of the algorithm, very good experimental results overall.

Weaknesses: Somewhat limited novelty.

Correctness: I have verified that the mathematical and algorithmic claims are correct.

Clarity: The paper is well written.

Relation to Prior Work: The comparison to the prior work is thorough and complete.

Reproducibility: Yes

Additional Feedback: The method proposed here is very simple while very effective, improved over several strong baselines in NeurIPS 2019 and ICLR 2020. Another benefit of this method is that it requires minimal computational overhead (only about 2x time compare to the original Cohen et al. 2019 paper, much faster than SmoothAdv and MACER). I would like to highlight that this method looks super natural and elegant: it only requires 2-3 lines of modification over MACER or Gaussian training, but improves a lot in terms of performance and efficiency. My only conservation is that this paper mostly follows the surrogate loss framework proposed by Zhai et al. 2020, which somewhat limits the novelty of this work.


Review 2

Summary and Contributions: The paper tackles the problem of learning a classifier robust to L2-ball attacks, by introducing a consistency regularizer that enforces consistency of the classifier's predictions with the smoothed classifier's predictions over the L2-ball. Performance of this new regularizer is evaluated extensively on several datasets, including ImageNet, where it outperforms competing approaches, in addition to being more computationally efficient.

Strengths: - Proposed regularizer is easy to incorporate and results in a method that is more efficient than competing approaches - Extensive evaluation on a range of datasets, in particular ImageNet, showing performance at scale - Ablation study comparing different design choices for the regularizer and sensitivity of performance to hyperparameters - Certifiable robustness is a topic of strong interest to the community and society in general

Weaknesses: - Experimental results seem to be on a single run. Given that some of the differences are not very large, ideally results from multiple runs are included to show the variance in metrics. - "Clean" accuracy and accuracy in the case of small perturbations is worse than MACER; this could compromise practical application where the clean accuracy is also important.

Correctness: Seems fine.

Clarity: The paper is generally easy to read, but there were a few parts that were confusing: - line 104: the "sufficient condition" didn't seem to be defined here, and was again referred to on line 107. What is this condition? - the last expectation in equation (6) - should this just be the probability associated with the indicator instead of an expectation

Relation to Prior Work: Yes, there is a section in the paper dedicated to this.

Reproducibility: Yes

Additional Feedback: Overall, this paper presents an efficient approach to training L2-robust models, that outperforms existing approaches in the large perturbation regime. While experiments could be improved with multiple runs, I thought they were extensive and included analyses of different design choices. Releasing code/models would help further improve reproducibility of the work. Additional comments: - Why does m have to be larger than 1? How does the method perform with m=1? - The analysis resulting in Figure 1 focuses on the log-probability gap, or logit-margin of the various classifiers. However, this is not the only factor contributing to robustness in the case of deep neural networks, which perform a highly non-linear mapping from inputs to logits; the distance to the decision boundary in input space (or input margin) is what we really care about, and is related to the logit-margin by the Lipschitzness of the mapping from input to logits; see Lipschitz-Margin Training: Scalable Certification of Perturbation Invariance for Deep Neural Networks, NeurIPS 2018 for a discussion. This analysis therefore doesn't really give the full picture of what is happening with the different algorithms. === Post Rebuttal Comments === I have read the rebuttal. Thank you for the clarifications.


Review 3

Summary and Contributions: This paper proposes a regularized term on the loss function to provide namely "Certified Robustness" for deep neural networks. The regularized term is simply a cross-entropy term and introduces relatively small overhead, but the performance increase is significant. ------------ Post rebuttal: most of my concerns are addressed and I think it is a good submission.

Strengths: 1) This paper is well-written and self-contained. 2) The major part of the theoretical grounding seems sound. 3) Experiment results are convincing and comprehensive.

Weaknesses: 1) Some notations seem not defined before use, and I have questions on Eq. 6 and Eq. 7 (see questions below). 2) The broader impact part seems missing some items provided by the NeurIPS template, e.g. I do not see answers to "Who may benefit from this research", "Who may be put at disadvantage from this research" and "What are the consequences of failure of the system". The broader impact part mighit need a full revision before publication.

Correctness: Can you clearly state the gap between Eq. 6 and Eq. 7? From my perspective, Eq. 6 means maximizing P_\delta(f(x+\delta)=\hat{f}(x)) and Eq. 7 is standard cross-entropy. However, P_\delta(f(x+\delta)=\hat{f}(x)) does not grow monotonically if the cross-entropy grows, since there are cases where models have different cross-entropy losses but the same P_\delta(f(x+\delta)=\hat{f}(x)) is achieved. The authors should state the gap more clearly between Eq. 6 and Eq. 7.

Clarity: Generally yes. I have the following questions/suggestions: 1) What does the bold 1 in Eq. 2 & Eq. 5 mean? This seems not a standard notation and should be explained. 2) It will make it clearer to add references to the legend of Fig. 1. I cannot realize that "consistency" is the proposed method when I reach Fig. 1.

Relation to Prior Work: Yes.

Reproducibility: Yes

Additional Feedback: Why SmoothAdv + Consistency is not considered in Table 2 for ImageNet dataset?


Review 4

Summary and Contributions: - Paper proposes a simple consistency regularization term added on a standard training scheme surprisingly improves the certified robustness of smoothed classifiers. - Owing to the simplicity of the method it allows faster training of robust classifiers. - Even though results are similar to other approaches, the speed up in training is a desirable property. - Observations show that the single hyper-parameter of the method is stable across multiple choices; showing the methods robustness/in-sensitivity to hyper-parameters.

Strengths: - Compared to other approaches, this offers significantly less training cost with fewer hyperparameters. - Using the proposed method with SmoothAdv [30] shows further improvement of the certified robustness (although it is significantly more expensive). This shows that the method can be combined with other methods in different ways and still show its benefits. - Results on a range of different experiments, including imagenet. - Experimental results are convincing showing at times the benefit of the consistency regularization, though the strengthes lie in that it speeds up the training process.

Weaknesses: - Would have been interesting to see the results on multiple models for the same dataset. It is not uncommon that the networks behave very differently given different network architectures.

Correctness: Yes

Clarity: Yes

Relation to Prior Work: Yes

Reproducibility: Yes

Additional Feedback:

[Author Response · NeurIPS 2020]

We sincerely thank all the reviewers for their thoughtful comments, efforts, and time. We are delighted that every
reviewers has left a positive impression on our research, particularly appreciating the simplicity and effectiveness of our
method, with thorough and strong experimental results. We respond to each comment one-by-one in what follows.

**(R1) Novelty.** We use the framework in [4] since it is a natural way to extend the previous natural-robust error
decomposition [5] for smoothed classifiers. We believe the novelty of our method is in its simplicity, and in how to
come up with this simplest form from the common principle of accuracy-robustness trade-off under the framework.

**(R2) Variance over multiple runs.** We observe ACR of a
training method is fairly robust to network initialization, e.g.,
as given in the right table: each value reports the mean and
standard deviation across 5 seeds. As another support, we
point out all the baselines considered in this paper [1, 3, 2, 4]
also report single-run results in their papers, possibly based
on observations similar to ours. Finally, we plan to publicly
release our code and models for better reproducibility.

| ACR (MNIST) | $\sigma = 0.25$ | $\sigma = 0.50$ | $\sigma = 1.00$ |
|---|---|---|---|
| Gaussian [1] | $0.9108_{\pm 0.0003}$ | $1.5581_{\pm 0.0016}$ | $1.6184_{\pm 0.0021}$ |
| **+ Consistency** | $\mathbf{0.9300}_{\pm \mathbf{0.0004}}$ | $\mathbf{1.6653}_{\pm \mathbf{0.0007}}$ | $\mathbf{1.7486}_{\pm \mathbf{0.0025}}$ |
| SmoothAdv [3] | $0.9315_{\pm 0.0001}$ | $1.6830_{\pm 0.0006}$ | $1.7706_{\pm 0.0019}$ |
| **+ Consistency** | $\underline{\mathbf{0.9323}}_{\pm \mathbf{0.0004}}$ | $\underline{\mathbf{1.6905}}_{\pm \mathbf{0.0002}}$ | $\underline{\mathbf{1.8087}}_{\pm \mathbf{0.0022}}$ |
| Stability [2] | $0.9152_{\pm 0.0007}$ | $1.5719_{\pm 0.0028}$ | $1.6341_{\pm 0.0018}$ |
| MACER [4] | $0.9201_{\pm 0.0006}$ | $1.5899_{\pm 0.0069}$ | $1.5950_{\pm 0.0051}$ |

**(R2) Clean accuracy is often worse than [4].** Our method can effectively explore the accuracy-robustness trade-off
[5] with $\lambda$. We expect the demands for higher clean accuracy could be compensated by using lower $\lambda$, e.g., our method
achieves better clean accuracy than MACER in Table 1 when $\lambda = 10$ on $\sigma = 0.50$, even with better ACR. Nevertheless,
it is remarkable that MACER sometimes achieve higher clean accuracy even than Gaussian, e.g., at $\sigma = 0.25$ of Table 1,
and we agree with **R2** that improving clean accuracy of smoothed classifier is also an important future direction.

**(R2) Logit margin vs. input margin.** Regarding Figure 1, it is important to notice that our focus is NOT the robustness
of the base classifiers $f$ tested, but the robustness of its *smoothed* counterparts $\hat{f}$. A key benefit of smoothed classifier is
that it elegantly transforms a matter of *input* margin on $f$ into that of *output* margin on $\hat{f}$: the robustness guarantee of
Cohen et al. [1] in Eq. 3 implies that one is enough to minimize $\mathbb{P}_\delta(f(x + \delta) \neq y)$ to improve the robustness of $\hat{f}$ at $x$,
which corresponds to the shaded areas in Figure 1. This can be also viewed in terms of the Lipschitzness: Salman et al.
[3] have shown that any Gaussian-smoothed classifier $\hat{f}$ has an explicit Lipschitz constant, leading to a simpler proof of
Eq. 3 [1]. We will incorporate the respective discussion in the final draft.

**(R2) "Sufficient condition"? The last $\mathbb{E}$ in Eq. 6? Why $m > 1$?** (*i*) The condition we refer is at L107: "$F(x + \delta)$
returns a constant output over $\delta$". This directly implies "Eq. 6 → 0", and consequently "the robust error of Eq. 5 → 0",
which is why we refer this as *sufficient* condition. (*ii*) Regarding Eq. 6, we remark the outer $\mathbb{E}$ is not over $\delta$, but over
$(x, y) \sim \mathcal{D}$ (as given in Eq. 5), thereby the last $\mathbb{E}$ of Eq. 6 cannot be discarded. (*iii*) Finally, our regularization requires
$m > 1$ to work, as the term would vanish if $m = 1$: with only a single sample, say $\delta_1$, $F(x + \delta_1)$ would be the best
estimation of $\hat{F}(x)$ in Eq. 7, and $L^{\mathtt{con}} = 0$ in this case. We will make all these points more clearer in the final draft.

**(R3) Eq. 6 → Eq. 7?** For a fixed $x$, the cross-entropy loss in Eq. 7 is a natural surrogate loss of the 0-1 risk
$\mathbb{E}_\delta[\mathbf{1}_{f(x+\delta) \neq \hat{f}(x)}] = \mathbb{P}_\delta(f(x + \delta) \neq \hat{f}(x))$, and this 0-1 risk minimizes the last upper bound in Eq. 6 when minimized
across $(x, y) \sim \mathcal{D}$. We also remark that this surrogate loss is *calibrated* [6, 5], i.e., minimizers of Eq. 7 are also
minimizers of the 0-1 risk (as mentioned in L104-110). We will clarify this point in the final draft.

**(R3) Suggestions for better clarity.** (*i*) We use $\mathbf{1}_A$ to denote the *indicator* random variable, formally defined by
$\mathbf{1}_A(\omega) = 1$ if $\omega \in A$, and 0 otherwise. We will specify this in the final draft. (*ii*) As suggested by **R3**, we will update
the legends in Figure 1 to better indicate our method. (*iii*) Also, we thank **R3** for a detailed assessment of the Broader
Impact statements. The final draft will include more discussions regarding the points made by **R3**.

**(R3) SmoothAdv + Consistency on ImageNet?** We conduct ImageNet experiments primarily on Gaussian training,
and report SmoothAdv results for a comparison. Conducting the suggested experiments with SmoothAdv would be
interesting, but currently we found it incurs too much costs to execute during the rebuttal period, e.g., ∼600 GPU hours
per single run. Nevertheless, we are willing to incorporate them in the final draft for thoroughness of our experiments.

**(R4) Other architectures.** In our experiments, all the architectures per dataset is exactly from the prior works [1, 3, 4]
for a fair comparison. Nevertheless, we agree with **R4** that the effect of architectures on smoothed classifiers is an
important question to explore, and we will include more results on other architectures, e.g., DenseNet, in the final draft.

[1] J. Cohen et al. Certified adversarial robustness via randomized smoothing. In *ICML*, 2019.
[2] B. Li et al. Certified adversarial robustness with additive noise. In *NeurIPS*, 2019.
[3] H. Salman et al. Provably robust deep learning via adversarially trained smoothed classifiers. In *NeurIPS*, 2019.
[4] R. Zhai et al. MACER: Attack-free and scalable robust training via maximizing certified radius. In *ICLR*, 2020.
[5] H. Zhang et al. Theoretically principled trade-off between robustness and accuracy. In *ICML*, 2019.
[6] B. Ávila Pires and C. Szepesvári. Multiclass classification calibration functions, 2016.


[Meta-Review · NeurIPS 2020]

Thank you for your submission to NeurIPS. All the reviewers and I agree that this paper presented a substantial contribution to the topic of certified robustness. The reviewers all notably appreciated the simplicity and strong performance of the proposed approach, and were unanimous is supporting acceptance.